# The Impact of Complement Genes on the Risk of Late-Onset Alzheimer’s Disease

**DOI:** 10.3390/genes12030443

**Published:** 2021-03-20

**Authors:** Sarah M. Carpanini, Janet C. Harwood, Emily Baker, Megan Torvell, Rebecca Sims, Julie Williams, B. Paul Morgan

**Affiliations:** 1UK Dementia Research Institute at Cardiff University, School of Medicine, Cardiff, CF24 4HQ, UK; CarpaniniS@Cardiff.ac.uk (S.M.C.); BakerEA@cardiff.ac.uk (E.B.); TorvellM@cardiff.ac.uk (M.T.); WilliamsJ@cardiff.ac.uk (J.W.); 2Division of Infection and Immunity, School of Medicine, Systems Immunity Research Institute, Cardiff University, Cardiff, CF14 4XN, UK; 3Division of Psychological Medicine and Clinical Neurosciences, School of Medicine, Cardiff University, Cardiff, CF24 4HQ, UK; HarwoodJC@cardiff.ac.uk (J.C.H.); SimsRC@cardiff.ac.uk (R.S.)

**Keywords:** complement, complement receptor 1, clusterin, late-onset Alzheimer’s disease, genetics, neuroinflammation

## Abstract

Late-onset Alzheimer’s disease (LOAD), the most common cause of dementia, and a huge global health challenge, is a neurodegenerative disease of uncertain aetiology. To deliver effective diagnostics and therapeutics, understanding the molecular basis of the disease is essential. Contemporary large genome-wide association studies (GWAS) have identified over seventy novel genetic susceptibility loci for LOAD. Most are implicated in microglial or inflammatory pathways, bringing inflammation to the fore as a candidate pathological pathway. Among the most significant GWAS hits are three complement genes: *CLU*, encoding the fluid-phase complement inhibitor clusterin; *CR1* encoding complement receptor 1 (CR1); and recently, *C1S* encoding the complement enzyme C1s. Complement activation is a critical driver of inflammation; changes in complement genes may impact risk by altering the inflammatory status in the brain. To assess complement gene association with LOAD risk, we manually created a comprehensive complement gene list and tested these in gene-set analysis with LOAD summary statistics. We confirmed associations of *CLU* and *CR1* genes with LOAD but showed no significant associations for the complement gene-set when excluding *CLU* and *CR1*. No significant association with other complement genes, including *C1S*, was seen in the IGAP dataset; however, these may emerge from larger datasets.

## 1. Introduction

Alzheimer’s disease (AD) is the most common cause of dementia in the elderly. Pathologically, AD is a chronic neurodegenerative disease underpinned by neuronal and synaptic loss, the accumulation of amyloid-β plaques, and neurofibrillary tangles composed of hyperphosphorylated tau. An important role for neuroinflammation has emerged in recent years. Evidence includes the presence of activated microglia in the brain innate immune cells, the presence of inflammatory markers, including complement proteins, in the brain, cerebrospinal fluid (CSF) and plasma, and the demonstration that chronic use of anti-inflammatory drugs may reduce disease incidence [1,2,3]. Perhaps the best evidence that inflammation may be involved in AD aetiology comes from genome-wide association studies (GWAS); many of the genes most strongly associated with AD risk are involved in inflammation and immunity. 

The first causative mutations for AD, identified over 25 years ago in the rare early-onset familial forms of AD, were in Amyloid precursor protein (*APP*), Presenilin 1 (*PSEN1*) and Presenilin 2 (*PSEN2*) genes [4,5,6]. APP, encoded by the *APP* gene, a broadly expressed transmembrane protein abundant in the brain, is sequentially cleaved by secretase enzymes. The precise cleavage patterns determine its propensity to seed Aβ plaques. The presenilin proteins PSEN1 and PSEN2 are both components of the γ-secretase complex and important in the function of this enzyme; mutations in the genes encoding these proteins impact the APP cleavage pathway. The identification of early-onset AD-associated mutations in these three genes underpins the amyloid cascade hypothesis whereby abnormal APP processing leading to Aβ plaque formation is considered the key underlying pathology associated with AD [7]. However, it is important to stress that these mutations are only relevant to early-onset familial AD which accounts for fewer than 1% of all AD cases. In late-onset Alzheimer’s disease (LOAD), accounting for the large majority of AD cases, the strongest genetic risk factor is the presence of the ε4 allele of the gene encoding Apolipoprotein E (ApoE); ε4 confers increased risk, while the most common allele, ε3, is considered neutral for AD, and ε2 has a minor protective effect [8,9,10,11]. Homozygosity for *APOE* ε4 confers an ~11-fold increased risk of LOAD compared to ε3 homozygotes. Precisely how these variants in *APOE* impact disease risk remains a subject of ongoing research. ApoE is a lipoprotein present in biological fluids; therefore, roles in lipid transport and membrane repair in the brain have been proposed [12]. 

Over the past decade, large GWAS have identified variants in more than 70 genetic loci that are associated with LOAD, implicating multiple and diverse biological pathways [13,14,15,16]. Notably, ~20% of the genes in LOAD risk loci encode proteins with roles in inflammation and immunity [14,17,18]; many of these are predominantly expressed in microglia, notably *TREM2, ABI3* and *PLCG2* [15,19]. From GWAS, it has been shown that three complement system genes are significantly associated with LOAD: *CLU*, *CR1*, and recently, *C1S* encoding the classical pathway enzyme C1s was added to this list [13,16,20]. *CLU* encodes clusterin, a multifunctional plasma protein that regulates the complement terminal pathway, and *CR1* encodes complement receptor 1 (CR1), a receptor for complement fragments and regulator of activation. These are both regulators of the complement cascade and provide the impetus for this analysis of complement genetics in LOAD. To test whether complement genes beyond *CLU* and *CR1* (both genome-wide significant (GWS) in the International Genomics of Alzheimer’s Project (IGAP) dataset) influence the risk of LOAD, we compiled a comprehensive complement gene-set containing only those genes that encoded proteins directly involved in complement activation, regulation, or recognition. Then, we undertook several methods of pathway analysis to test whether additional genes within the complement gene-set were associated with LOAD risk. 

## 2. Materials and Methods

### 2.1. Complement Genes and Gene Exclusion Analyses in LOAD

In order to understand the genetics of the complement pathway in AD, we compiled a comprehensive gene-set comprising all complement genes and associated regulators and receptors. Genes were selected for inclusion based upon known biological relevance to the complement system rather than by using often inaccurate annotations in public databases. The resultant complement gene-set contained 56 genes, subdivided into their relevant functional groups (Table 1). 

### 2.2. AD Summary Statistics 

This study utilised summary statistics from the International Genomics of Alzheimer’s Project (IGAP). IGAP is a large three-stage study based upon GWAS on individuals of European ancestry. In stage 1, IGAP used genotyped and imputed data on 11,480,632 single nucleotide polymorphisms (SNPs) to meta-analyse GWAS datasets consisting of 21,982 Alzheimer’s disease cases and 41,944 cognitively normal controls from four consortia: the Alzheimer Disease Genetics Consortium (ADGC); the European Alzheimer’s disease Initiative (EADI); the Cohorts for Heart and Aging Research in Genomic Epidemiology Consortium (CHARGE); and the Genetic and Environmental Risk in AD Consortium Genetic and Environmental Risk in AD/Defining Genetic, Polygenic and Environmental Risk for Alzheimer’s Disease Consortium (GERAD/PERADES). In stage 2, 11,632 SNPs were genotyped and tested for association in an independent set of 8362 Alzheimer’s disease cases and 10,483 controls. Meta-analyses of variants selected for analysis in stage 3A (*n* = 11,666) or stage 3B (*n* = 30,511) samples brought the final sample to 35,274 clinical and autopsy-documented Alzheimer’s disease cases and 59,163 controls. 

Gene-set analysis was performed using the complement gene-set and stage 1 summary statistics from the International Genomics of Alzheimer’s Project [14]. The individual and combined effects of the genome-wide significant (GWS) genes *CLU* and *CR1* within the complement gene-set were investigated by removing these genes individually and together. We utilised the most up-to-date publicly available GWAS dataset at the time of writing [14], and calculated the complement gene-set *p*-values when including and excluding those loci that reached genome-wide significance in the IGAP dataset. The recently identified LOAD-associated *C1S* variant [13] does not show genome-wide statistical significance in the IGAP dataset; and therefore was not removed in the gene-set analysis. Complement gene-sets were tested for enrichment using the IGAP stage 1 summary statistics [14] in MAGMA version 1.06 [21]. Summary statistics were filtered for common variants (MAF ≥ 0.01) and all indels and merged deletions were removed; 8,608,484 SNPs were analysed. Genes were annotated using reference data files from the European population of Phase 3 of 1000 Genomes, human genome Build 37 using a window of 35 kb upstream and 10 kb downstream of each gene [22]. Ten thousand permutations were used to estimate *p*-values, corrected for multiple testing using the family-wise error rate (FWER). Gene-sets with a FWER-corrected *p*-value < 0.05 under the “mean” model for estimating gene-level associations were reported as significant.

### 2.3. Complement Risk Score Analysis

A complement risk score combining the effects of all SNPs in the complement gene-set was produced. POLARIS [23] was used to compute risk scores in GERAD-genotyped data (3332 cases, 9832 controls) using SNP effect sizes from IGAP stage 1 summary statistics [14,16,20] (excluding GERAD subjects). Linkage disequilibrium (LD) was estimated from the GERAD data, and POLARIS was used to adjust the scores for LD between SNPs. The overall association of the complement gene-set with LOAD was determined using a logistic regression model, adjusting for population covariates, age, and sex. The logistic regression model included the baseline polygenic risk scores for all SNPs in the model, thereby testing for any association beyond the baseline polygenic effect. 

Data used in the preparation of this article were obtained from the Genetic and Environmental Risk for Alzheimer’s disease (GERAD) Consortium. The imputed GERAD sample comprised 3177 AD cases and 7277 controls with available age and gender data. Cases and elderly screened controls were recruited by the Medical Research Council (MRC) Genetic Resource for AD (Cardiff University; Institute of Psychiatry, London; Cambridge University; Trinity College Dublin), the Alzheimer’s Research UK (ARUK) Collaboration (University of Nottingham; University of Manchester; University of Southampton; University of Bristol; Queen’s University Belfast; the Oxford Project to Investigate Memory and Ageing (OPTIMA), Oxford University); Washington University, St Louis, United States; MRC PRION Unit, University College London; London and the South East Region AD project (LASER-AD), University College London; Competence Network of Dementia (CND) and Department of Psychiatry, University of Bonn, Germany; the National Institute of Mental Health (NIMH) AD Genetics Initiative. A total of 6129 population controls were drawn from large existing cohorts with available GWAS data, including the 1958 British Birth Cohort (1958BC) (http://www.b58cgene.sgul.ac.uk, accessed on 15 March 2021), the KORA F4 Study, and the Heinz Nixdorf Recall Study. All AD cases met criteria for either probable (NINCDS-ADRDA, DSM-IV) or definite (CERAD) AD. All elderly controls were screened for dementia using the MMSE or ADAS-cog and were determined to be free from dementia at neuropathological examination or had a Braak score of 2.5 or lower. Genotypes from all cases and 4617 controls were previously included in the AD GWAS by Harold and colleagues (2009) [20]. Genotypes for the remaining population controls were obtained from WTCCC2. Imputation of the dataset was performed using IMPUTE2 and the 1000 genomes (http://www.1000genomes.org/, accessed on 15 March 2021) Dec2010 reference panel (NCBI build 37.1). 

### 2.4. Likelihood Ratio Analysis

A likelihood ratio test was used to estimate how much of the complement gene-set effect on LOAD risk was contributed by *CLU* and *CR1*, and to test whether there were residual polygenic effects of the remaining genes from the complement gene-set. The effects of *CLU* and *CR1* were estimated using a risk score combining all SNPs in the gene, produced using POLARIS in order to correct for LD. Likelihood ratio tests were used to compare individual models containing SNPs in *CLU* and *CR1* and models containing the combined risk conferred by SNPs in the rest of the complement gene-set. 

## 3. Results

### 3.1. MAGMA Analysis Reveals the Impact of Individual Complement Genes

From the MAGMA gene-set analysis, the complement gene-set comprising all 56 genes was significantly associated with LOAD (*p* = 0.011) (Table 2). When the GWAS-significant genes *CLU* and *CR1* were excluded individually from the gene-set, the complement-minus-*CLU* gene-set was not significant (*p* = 0.057), while the complement-minus-*CR1* gene-set was significant (*p* = 0.048). As *CR1* and *CR1L* are located next to each other on chromosome 1, and linkage disequilibrium extends between the two genes, we excluded the *CR1/CR1L* locus from the gene-set. This gene-set was not significant (*p* = 0.082). The gene set in which both *CLU* and *CR1* were excluded from the complement gene-set was not significantly associated with LOAD (*p* = 0.170). The signal in the gene-set where *CR1L*, *CLU* and *CR1* were excluded was reduced compared with the signal derived from the gene-sets in which *CLU* and *CR1* were removed (Table 2). Taken together, these results suggest that the LOAD association signal in the complement gene-set is predominantly driven by *CLU* and *CR1.* Given the physical distance between *CR1* and *CR1L*, the use of extended gene boundaries and that linkage disequilibrium extends across both genes, we cannot resolve the signal between these two genes in the gene set analysis. Hence, we cannot confirm any independent contribution from *CR1L*.

Table 2 displays the results from the MAGMA analysis. Gene-sets were corrected for multiple testing using the family-wise error rate (FWER). The complement gene-set is significant (*p* = 0.011), but this effect is lost when *CLU* and *CR1* are excluded from the gene-set (*p* = 0.170). *CLU* has the largest impact in the complement set, and the association with AD is predominantly driven by *CLU* and *CR1*. 

### 3.2. Risk Score Analysis Supports the Impact of Complement Genes

To further explore the impact of complement genes on LOAD risk, we adopted a polygenic approach. We first applied risk score analysis to the dataset, then used logistic regression to explore the association between LOAD and complement gene-set risk scores in GERAD individuals (Table 3). The complement gene-set as a whole was strongly associated with AD in this analysis (*p* = 0.003). Removal of *CLU* from the gene-set caused the largest reduction in significance (*p* = 0.003 vs. *p* = 0.053). Removal of *CR1*, or the *CR1/CR1L* locus had minimal impact on the significance of association in the gene-set, although when *CLU, CR1* and *CR1L* were eliminated, the significance was further reduced compared to the elimination of *CLU* alone (*p* = 0.148 vs. *p* = 0.053) (Table 3). These gene elimination analyses demonstrated that *CLU* and *CR1* were the major contributors to the risk of LOAD in the complement gene-set; however, the polygenic approach revealed that *CLU* was by far the more significant of these. In these data, the *CLU* gene shows a stronger association compared to *CR1* (*p* = 1.03 × 10^−5^ and *p* = 1.5 × 10^−3^, respectively). The joint association of *CLU* and *CR1* is stronger still (*p* = 3.88 × 10^−7^), showing that *CLU* and *CR1* are both independently associated with AD.

Table 3 displays the results from the risk score analysis; the overall complement risk score shows an association with AD (*p* = 0.003). *CLU* explains the majority of this signal.

### 3.3. Likelihood Ratio Analysis Confirms No Significant Impact of Other Complement Genes

We next tested complement gene-set effects using likelihood ratio analyses. Models in which *CLU, CR1* and *CR1/CR1L* were removed individually, showed significant residual impact in the gene-set (*p* = 0.0136; *p* = 0.0091; *p* = 0.0063 respectively); after removal of *CR1* and *CLU* or *CR1*, *CLU* and *CR1L*, there was no significant residual impact in the gene-set, demonstrating that there was no significant polygenic effect of the remaining complement genes in the datasets used (Table 4). These results further support the conclusion that the complement gene-set association with LOAD is driven predominantly by *CLU* and *CR1*, but with no significant contribution from other complement gene-set members (*p* = 0.1457; Table 4).

Table 4 shows the results from these likelihood ratio tests comparing models containing SNPs in *CLU, CR1* and *CR1/CR1L* only and models containing the combined risk in SNPs in the remaining complement genes. The *p*-values demonstrate whether the remaining genes in the complement explain any additional variation. These results further support the conclusion that the complement gene-set impact on LOAD risk is predominantly driven by *CLU* and *CR1*. 

## 4. Discussion

The first evidence implicating the complement system in LOAD came from immunostaining of post-mortem brain tissue. Complement components and activation products, notably C1q, C4b, C3b/iC3b and the membrane attack complex, were present and co-localised with amyloid plaques and neurofibrillary tangles in the AD brain [24,25,26,27]. C3 fragments were shown to opsonise amyloid for phagocytosis by microglia in the brain and facilitate transport on erythrocytes to the liver [28]. Complement activation is critically involved in synaptic pruning both in development and in diseases such as AD [29,30,31,32]. In AD mouse models, back-crossing to complement deficiencies has supported the critical role of complements in neuroinflammation and synapse loss [30,33]. The presence of complement activation biomarkers in CSF and/or plasma in LOAD suggested that complement dysregulation occurs early in the disease [2]. The demonstration that complement genes associated with LOAD provided compelling evidence that the complement was a driver of disease rather than a secondary event [13,14,16]. 

To further investigate the roles of complement genes in the risk of LOAD, we compiled a comprehensive complement gene-set and used a polygenic approach to identify genes contributing to AD risk. We have demonstrated that the signal for the association of the complement gene-set with LOAD is explained by the GWS genes *CLU* and *CR1*, and not by other complement genes tested here. This finding was unexpected. Based on knowledge from other chronic inflammatory diseases, we had hypothesised that many complement genes might influence LOAD risk. For example, in age-related macular degeneration (AMD), a retinal disease clinically and pathologically linked to LOAD, genes encoding complement components C2, C3, FB and C9, and regulators FH and FHR4, all contribute to risk [34]. Indeed, the demonstration that multiple complement genes can collaborate to cause dysregulation and disease informed the concept of the “complotype”, the set of complement gene variants inherited by an individual that dictates complement activity and disease risk [35]. The genetic associations in these other chronic inflammatory diseases influence systemic or local complement regulation and/or amplification of activation; these in turn cause complement dysregulation that drives inflammation. Our demonstration that the complement genetic signature in LOAD is restricted to the genes encoding clusterin and CR1 suggests that complement dysregulation is not critical in the disease process. However, it should be noted that this finding is dependent on the dataset being investigated. At the time of writing, we utilised the largest publicly available AD GWAS dataset [14]. A recent study by the European AD Biobank, currently a preprint, reported an LOAD GWAS-significant association with the complement gene *C1S* [13]; this suggests that larger datasets and different analytical methods may implicate other complement genes and further elucidate roles of the complement system in LOAD. Additionally, because of the highly repetitive nature of a number of the complement loci, for example, the regulators of complement activation (RCA) clusters on chromosome 1 [36], many complement genes may be hidden from standard sequencing technologies; the application of emerging long-range sequencing methods may reveal additional genetic variation in complement genes linked to LOAD missed in current GWAS and whole exome/genome sequencing studies using short read sequencing technologies [37].

Of the complement genes tested here, *CLU* and *CR1* were significantly associated with LOAD through multiple analytical approaches. Clusterin is a multi-functional plasma protein; its role in the complement system is to restrict fluid-phase membrane attack pathway activation [38]; however, beyond the complement system, clusterin functions as an extracellular chaperone protein, is involved in oxidative stress and cell survival/cell death pathways, and functions as an apolipoprotein in lipid transport [38,39,40,41]. Any one or several of these functions might underpin the association with LOAD. Four SNPs in *CLU,* all intronic and in LD, have been associated with increased LOAD risk (rs11136000, rs2279590, rs9331888 and rs9331896) [16,20]; evidence to date suggests that these SNPs impact clusterin synthesis, and hence, plasma clusterin levels. CR1 is a membrane-bound receptor for complement components (C1q, MBL) and fragments (C3b, C4b). The primary function of CR1 is as a receptor for C3b/C4b-opsonised immune complexes. CR1 on erythrocytes sequesters immune complexes and transports them to disposal sites, while CR1 on phagocytic cells binds opsonised immune complexes and processes them for elimination via phagocytosis. This latter activity requires a second function of CR1, its cofactor activity for factor I cleavage of C3b to iC3b the ligand for the phagocytic receptor CR3. The biological relevance of the C1q/MBL binding functions of CR1 are unclear. The human *CR1* gene is located in the RCA gene cluster on chromosome 1 (1q32); duplications and deletions in this highly repetitive gene generate multiple isoforms via copy number variation (CNV). The most common variant, CR1*1 (allele frequency 0.87) comprises 30 tandem repeats of 60–70 amino acid units called short consensus repeats (SCRs), which are in turn grouped in four homologous sets of seven termed long homologous repeats (LHRs), each a separate C3b/C4b binding unit. The second most common variant CR1*2 (allele frequency 0.11) is identical to CR1*1 except for the acquisition of an additional LHR, a “gain-of-function”; this variant increases risk for LOAD by up to 30%, although precisely how is unclear [14,16,42,43,44]. It has been suggested that the CR1*2 variant is associated with lower CR1 expression on erythrocytes, reducing the efficiency of peripheral immune complex handling and impacting amyloid clearance from the brain [45,46]. 

Our original analysis suggested that some of the signal from the complement gene-set might be attributable to the *CR1L* gene. However, *CR1L* is immediately adjacent to *CR1* and the SNP signals cannot be resolved, so it is not possible to ascribe an independent signal to *CR1L* in this analysis. *CR1L* encodes a C4b-binding protein comprising 13 SCRs, expressed predominantly in haematopoietic tissues [47,48]. Its physiological role is unknown, and evidence mechanistically linking it to LOAD is absent. 

## 5. Conclusions

Taken together, our findings confirm the strong genetic association of the complement genes *CR1* and *CLU* with LOAD and that there is no statistically significant association signal for other complement genes apparent in the dataset used for the analysis. CR1 and clusterin are important regulators of the complement pathway, suggesting that its dysregulation is important in LOAD. The recent GWAS association of *C1S* with LOAD demonstrates the potential for missing associations in this complex gene-set and raises the possibility that other loci may be missed by current large-scale genotyping and short-read sequencing technologies. Application of long read sequencing technologies could significantly alter the current landscape of complement system genetics in relation to LOAD risk.

## Figures and Tables

**Table 1 genes-12-00443-t001:** Complement gene list including all complement genes and associated regulators and receptors. Genes are sub-divided according to pathway; either classical, lectin, amplification loop or terminal and whether they are complement genes or associated regulators/receptors.

Pathway	HGNC Gene Name	Entrez Gene ID	HGNC Full Gene Name
Classical	*C1QA*	712	complement C1q A chain
Classical	*C1QB*	713	complement C1q B chain
Classical	*C1QC*	714	complement C1q C chain
Classical	*C1R*	715	complement C1r
Classical	*C1S*	716	complement C1s
Classical/Lectin	*C2*	717	complement C2
Classical/Lectin	*C4A*	720	complement C4A (Rodgers blood group)
Classical/Lectin	*C4B*	721	complement C4B (Chido blood group)
Lectin	*FCN1*	2219	ficolin 1
Lectin	*FCN2*	2220	ficolin 2
Lectin	*FCN3*	8547	ficolin 3
Lectin	*MASP1*	5648	mannan binding lectin serine peptidase 1
Lectin	*MASP2*	10747	mannan binding lectin serine peptidase 2
Lectin	*MBL2*	4153	mannose binding lectin 2
Amplification loop	*CFB*	629	complement factor B
Amplification loop	*CFD*	1675	complement factor D
Classical/Lectin/ Amplification loop	*C3*	718	complement C3
Terminal	*C5*	727	complement C5
Terminal	*C6*	729	complement C6
Terminal	*C7*	730	complement C7
Terminal	*C8A*	731	complement C8 α chain
Terminal	*C8B*	732	complement C8 β chain
Terminal	*C8G*	733	complement C8 γ chain
Terminal	*C9*	735	complement C9
Regulator/Receptor	*C1QBP*	708	complement C1q binding protein
Regulator/Receptor	*C3AR1*	719	complement C3a receptor 1
Regulator/Receptor	*C4BPA*	722	complement component 4 binding protein α
Regulator/Receptor	*C4BPB*	725	complement component 4 binding protein β
Regulator/Receptor	*C5AR1*	728	complement C5a receptor 1
Regulator/Receptor	*C5AR2*	27202	complement component 5a receptor 2
Regulator/Receptor	*CD46*	4179	CD46 molecule
Regulator/Receptor	*CD55*	1604	CD55 molecule (Cromer blood group )
Regulator/Receptor	*CD59*	966	CD59 molecule
Regulator/Receptor	*CFH*	3075	complement factor H
Regulator/Receptor	*CFHR1*	3078	complement factor H related 1
Regulator/Receptor	*CFHR2*	3080	complement factor H related 2
Regulator/Receptor	*CFHR3*	10878	complement factor H related 3
Regulator/Receptor	*CFHR4*	10877	complement factor H related 4
Regulator/Receptor	*CFHR5*	81494	complement factor H related 5
Regulator/Receptor	*CFI*	3426	complement factor I
Regulator/Receptor	*CFP*	5199	complement factor properdin
Regulator/Receptor	*CLU*	1191	clusterin
Regulator/Receptor	*CR1*	1378	complement C3b/C4b receptor 1 (Knops blood group)
Regulator/Receptor	*CR2*	1380	complement C3d receptor 2
Regulator/Receptor	*CSMD1*	64478	CUB and Sushi multiple domains 1
Regulator/Receptor	*ITGAM*	3684	integrin subunit α M
Regulator/Receptor	*ITGAX*	3687	integrin subunit α X
Regulator/Receptor	*SERPING1*	710	serpin family G member 1
Regulator/Receptor	*VTN*	7448	Vitronectin
Regulator/Receptor	*CD93*	22918	C1q receptor phagocytosis
Complement-like	*C1QL1*	10882	complement C1q-like 1
Complement-like	*C1QL2*	165257	complement C1q-like 2
Complement-like	*C1QL3*	389941	complement C1q-like 3
Complement-like	*C1QL4*	338761	complement C1q-like 4
Complement-like	*C1RL*	51279	complement C1r subcomponent-like
Complement-like	*CR1L*	1379	complement C3b/C4b receptor 1-like

**Table 2 genes-12-00443-t002:** Complement gene-set analysis.

Gene-Set	Ngenes	OR	95% CI	*p*	*p* FWER
Complement Genes	56	1.402	[1.068, 1.841]	0.008	0.011
Complement Genes Minus *CLU*	55	1.278	[0.969, 1.684]	0.041	0.057
Complement Genes Minus *CR1*	55	1.288	[0.981, 1.691]	0.034	0.048
Complement Genes Minus *CLU*, *CR1*	54	1.172	[0.891, 1.542]	0.129	0.170
Complement Genes Minus *CR1*, *CR1L*	54	1.244	[0.943, 1.639]	0.061	0.082
Complement Genes Minus *CLU*, *CR1*, *CR1L*	53	1.127	[0.854, 1.489]	0.199	0.246

**Table 3 genes-12-00443-t003:** Association between Alzheimer’s disease (AD) and complement gene-set risk score.

Gene-Set	Ngenes	OR	95% CI	*p*
Complement Genes	56	1.090	[1.028, 1.156]	0.003
Complement Genes Minus *CLU*	55	1.059	[0.998, 1.123]	0.053
Complement Genes Minus *CR1*	55	1.089	[1.027, 1.155]	0.004
Complement Genes Minus *CLU*, *CR1*	54	1.058	[0.997, 1.122]	0.059
Complement Genes Minus *CR1*, *CR1L*	54	1.077	[1.015, 1.142]	0.013
Complement Genes Minus *CLU*, *CR1*, *CR1L*	53	1.044	[0.984, 1.107]	0.148

**Table 4 genes-12-00443-t004:** Likelihood ratio test (LRT) comparing gene-set risk scores.

Models Compared	LRT *p*-Value
(1) *CLU* (2) *CLU* + Complement_minus_*CLU*	0.0136
(1) *CR1* (2) *CR1* + Complement_minus_*CR1*	0.0091
(1) *CLU* + *CR1* (2) *CLU* + *CR1* + Complement_minus_*CLU*_*CR1*	0.1457
(1) *CR1 + CR1L* (2) *CR1 + CR1L* + Complement_minus_*CR1_CR1L*	0.0063
(1) *CLU* + *CR1* + *CR1L* (2) *CLU* + *CR1* + *CR1L* + Complement_minus_*CLU*_*CR1*_*CR1L*	0.1145

## Data Availability

The GERAD data is available by request (GERADConsortium@cf.ac.uk) and Kunkle data is available to download (https://www.niagads.org/datasets/ng00075 accessed on 15 March 2021).

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
