# Peer review of "The Impact of Complement Genes on the Risk of Late-Onset Alzheimer’s Disease"

_genes, 2021, doi:10.3390/genes12030443_

Round 1
Reviewer 1 Report
This is informative and well-presented genetic study of late onset of Alzheimer Disease (LOAD), which used a polygenic approach and analysed up to 56 complement genes. The detected signal for the association with LOAD is mainly driven by two known LOAD genes (CLU and CR1).
There are only a few minor comments to be addressed.
- Re: “Genes were selected for inclusion based upon known biological relevance to the complement system rather than by using often inaccurate annotations in public databases”. It is not clear what is the actual strategy for the manual selection of 56 genes of interest in a situation when public databases were not used.
- What is the p-value if to use CLU and CR1 together vs the signal for each CLU or CR1 genes?
- There are repeats for abbreviation (eg, IGAP).
Author Response
The authors thank the reviewer for their prompt review and helpful comments. We are pleased the reviewer found the manuscript informative and well-presented. We have addressed each response one-by-one below.
There are only a few minor comments to be addressed.
1) Re: “Genes were selected for inclusion based upon known biological relevance to the complement system rather than by using often inaccurate annotations in public databases”. It is not clear what is the actual strategy for the manual selection of 56 genes of interest in a situation when public databases were not used.
Although automated pathway identification algorithms have proved valuable in many contexts we have been disappointed with them in the complement system where some complement genes are missed and several non-complement genes included. To address this we have used our own expertise and knowledge of the complement system to manually select the 56 genes of interest (and details on pathway and whether a complement component gene or a gene encoding a complement regulator/receptor). The senior author, a complement biologist of 30 years’ experience, brings considerable expertise to this task. The resultant gene list is comprehensive and accurately reflects the system.
2) What is the p-value if to use CLU and CR1 together vs the signal for each CLU or CR1 genes?
Thank you for highlighting this. We have added two sentences (lines 218-221) to define this.
“In these data, the CLU gene shows a stronger association compared to CR1 (p=1.03x10-5 and p=1.5x10-3 respectively). The joint association of CLU and CR1 is stronger still (p=3.88x10-7), strongly implying that CLU and CR1 are both independently associated with AD.”
3) There are repeats for abbreviation (eg, IGAP).
Thank you for pointing this out, the abbreviation repeats for IGAP, GWAS and LOAD have now been removed; each is defined once and the abbreviation used consistently thereafter.
Reviewer 2 Report
Minor comments:
1) In the regression table (tables 2 and 3) there is no need for SE (standard error?). However, I would include the odds ratios with their 95% confidence intervals.
2) In the Discussion there should be a most detailed discussion on the mechanisms linking complement, neuroinflammation, amyloid and neurodegeneration in AD.
Author Response
The authors thank the reviewer for their prompt review and helpful comments on where the manuscript can be improved. We have addressed each response individually below.
1) In the regression table (tables 2 and 3) there is no need for SE (standard error?). However, I would include the odds ratios with their 95% confidence intervals.
We have removed the SE in tables 2 and 3 and have included the OR and 95% CI as suggested.
2) In the Discussion there should be a most detailed discussion on the mechanisms linking complement, neuroinflammation, amyloid and neurodegeneration in AD.
We have added more detail in the first paragraph of the discussion.